# Size of Craniectomy Predicts Approach-Related Shear Bleeding in Poor-Grade Subarachnoid Hemorrhage

**DOI:** 10.3390/brainsci13030371

**Published:** 2023-02-21

**Authors:** Martin Vychopen, Johannes Wach, Tim Lampmann, Harun Asoglu, Hartmut Vatter, Erdem Güresir

**Affiliations:** 1Department of Neurosurgery, University Hospital Leipzig, 04103 Leipzig, Germany; 2Department of Neurosurgery, University Hospital Bonn, 53127 Bonn, Germany

**Keywords:** subarachnoid hemorrhage, decompressive craniectomy, shear bleeding

## Abstract

Decompressive craniectomy is an option to decrease elevated intracranial pressure in poor-grade aneurysmal subarachnoid hemorrhage (SAH) patients. The aim of the present study was to analyze the size of the bone flap according to approach-related complications in patients with poor-grade SAH. We retrospectively analyzed poor-grade SAH patients (WFNS 4 and 5) who underwent aneurysm clipping and craniectomy (DC or ommitance of bone flap reinsertion). Postoperative CT scans were analyzed for approach-related tissue injury at the margin of the craniectomy (shear bleeding). The size of the bone flap was calculated using the De Bonis equation. Between 01/2012 and 01/2020, 67 poor-grade SAH patients underwent clipping and craniectomy at our institution. We found 14 patients with new shear bleeding lesion in postoperative CT scan. In patients with shear bleeding, the size of the bone flap was significantly smaller compared to patients without shear bleeding (102.1 ± 45.2 cm^2^ vs. 150.8 ± 37.43 cm^2^, *p* > 0.0001). However, we found no difference in mortality rates (10/14 vs. 23/53, *p* = 0.07) or number of implanted VP shunts (2/14 vs. 18/53, *p* = 0.2). We found no difference regarding modified Rankin Scale (mRS) 6 months postoperatively. In poor-grade aneurysmal SAH, the initial planning of DC—if deemed necessary —and enlargement of the flap size seems to decrease the rate of postoperatively developed shear bleeding lesions.

## 1. Introduction

Decompressive craniectomy (DC) is a method for treatment of raised intracranial pressure (ICP). In patients with subarachnoid hemorrhage (SAH), many studies have reported the benefit of DC in case of ICP elevation due to brain swelling/cerebral vasospasm [1,2,3]. For patients with poor-grade SAH (WFNS 4 and 5), retrospective studies suggest that early DC can reduce mortality in some selected cases [4]. Buschmann et al. proposed that more than half of the patients with elevated ICP after SAH achieve good functional outcomes [5]. An ongoing trial PICASSO (Primäre dekompressive Kraniektomie bei aneurysmatischer Subarachnoidalblutung) [6] aims to evaluate the effect of the primary DC in poor-grade SAH patients in Germany in prospective multicenter setting.

In such cases, the size of the DC may be a crucial factor in determining the outcome. In contemporary literature, there is still no clear consensus regarding the size or the exact indication criteria in patients with SAH. Presented retrospective cohorts of Holsgrove et al. [7] found no relation between size of the DC and the outcome. On the other hand, a clear demonstration between ICP reduction and DC in patients with SAH was proven in study of Schimmer et al. [8]. Retrospectively, larger DC size seems to prevent the development of shear bleeding at the margin of the craniectomy [9].

Other authors suggest that increased size of the DCs does not provide greater protection against immediate surgical complications such as shear stress and cerebral herniation and that carefully performed temporobasal decompression decreases the intracranial pressure sufficiently even if the maximal size of the DC is not reached [10]. To date, the best agreement on the size of the decompression was reached in patients with traumatic brain injury, where a size of at least 12 × 15 cm is recommended to reach sufficient decompression [11].

For patients with poor-grade SAH indicated for aneurysm clipping, primary decompression might not be considered. In case of secondary bone flap ommitance, the initial craniotomy may not be large enough to provide sufficient ICP control and secondary enlargement of initially to small craniotomy might be necessary [12]. In case of persistent intracranial hypertension after the initial DC, even a bilateral decompression was suggested to improve the outcome in selected patients [13]. The aim of this study is to analyze the size of the DC and its effect on approach-related complications in patients undergoing emergent clipping and decompressive craniectomy due to aneurysm-related SAH.

## 2. Materials and Methods

We retrospectively searched our institutional SAH database for patients with aneurysm-related SAH who underwent clipping and DC/ommitance of the bone flap reinsertion.

All patients suspected of having poor-grade SAH received computer tomography (CT) scan on admission. Patients with positive CT findings underwent CT angiography and digital subtraction angiography (DSA) to identify the cause of the bleeding. Subsequently, multidisciplinary decision for clipping was made by each patient on individual basis considering the location and shape of the aneurysm, clinical status, and presence of acute signs of elevated intracranial pressure. All patients with acute hydrocephalus underwent placement of external ventricular drain prior to aneurysm treatment. Additionally, we retrospectively assessed and evaluated PRESSURE score [2].

Clipping was performed in standardized microsurgical procedure. The shaved head of the patient was fixated in Mayfield skull clamp. The size and anatomical margin of the resulting bony decompression are illustrated in Figure 1. An osteoclastic temporobasal decompression was performed by all patients.

Intraoperative, somatosensory and motor-evoked potentials were used for electrophysiological monitoring. After placing the clip, indocyanine green (ICG) video angiography was performed routinely to rule out the clip-induced stenosis or aneurysm residual.

All patients were postoperatively admitted to the specialized neuro-intensive-care unit. Computer tomography scans were obtained 24 h after decompression and clipping to assess the possible approach/procedure-related complications.

The size of the decompression was measured in postoperatively performed CT scan according to the DeBonis equation [14]. Moreover, the CT scans were examined for the presence of postoperative complications such as shear bleeding [9]. According to those findings, the patients were divided into two groups: patients with approach-related shear bleeding vs. patient without approach-related complications.

The neurological outcome was assessed according to modified Rankin Scale (mRS) 6 months postoperatively.

The statistical analysis of the data was performed with SPSS ICM Version 25. We used Fisher´s exact test to perform the univariate analysis. The size of the DC were compared between the groups with the use of one way ANOVA test. For the graphical representation of the data, a violin plot was made using GraphPad PRISM 9 for Windows. Finally, point biserial correlation coefficient was calculated. All results with *p* value < 0.05 were considered to be significant.

## 3. Results

We identified 88 patients (60 females) with aneurysmatic SAH who underwent clipping and decompressive craniectomy between 02/2012 and 12/2020. Excluded were patients treated with coil embolization and patients where only craniotomy was performed with subsequent reinsertion of the bone flap after aneurysm clipping. The mean age of the patients was 55.04 ± 13.53 years. The distribution according to WFNS was 10 patients with SAH WFNS 2, 11 patients with WFNS 3, 16 patients with SAH WFNS 4, and 51 patients with SAH WFNS 5. According to the Fisher scale, we found two patients with Fisher 2, 82 Patients with Fisher 3 and 4 patients with Fisher 4 bleeding. For detailed information, see the patient characteristics table (Table 1 and Table 2).

The analysis showed that patients with shear bleeding lesions had significantly smaller craniectomy surface compared to those without shear bleeding (102.1 ± 45.2 cm^2^ vs. 150.8 ± 37.43 cm^2^, *p* < 0.0001). We found no shear bleeding lesions among the patients with good-grade SAH (WFNS 2 and 3, craniectomy surface of 164.1 ± 35.3 cm^2^). We found no difference in baseline PRESSURE score between poor-grade patients with and without shear bleeding (6 (IQR 4.5–7) vs. 6 (IQR 5–7)).

We created violin plots to demonstrate a statistically significant difference in size of the DC between poor-grade SAH patients with and without shear bleeding. We saw no difference between poor-grade SAH patients without shear bleeding and good-grade SAH patients. For detailed information, see Figure 2.

Most of the shear bleedings were localized at the dorsomedial boundary of the craniectomy. We found only one temporobasal lesion. For detailed information on DC size and bleeding localization, see Figure 1 and Figure 3.

Finally, point biserial correlation coefficient (−0.46) showed that larger DC size has middle strong negative correlation with the incidence of shear bleeding lesions. We found no difference in the outcome among the groups (12/14 vs. 8/53 patients with mRS < 4; *p* > 0.99). The distribution of patients with mRS = 6 did not differ among the groups (10/14 vs. 23/53, *p* = 0.07). Furthermore, we saw no difference in number of implanted ventriculo peritoneal shunts (2/14 vs. 18/53; *p* = 0.2). For details on the outcomes, see Table 3.

## 4. Discussion

We performed a retrospective analysis of the bone flap size by the patients undergoing clipping and decompressive craniectomy due to aneurysmal SAH. Patients with shear bleeding injury showed significantly smaller size of the bone flap compared to patients without approach-related injuries. We found no approach-related injuries in patients with good-grade SAH. We found no significant difference in outcome between the groups but found a higher mortality trend in poor-grade patients with smaller DC size.

To date, there is still no agreement about the exact size and margin of the DC. The recommendation given on the size of the DC so far is the minimum of 12 × 15 cm large bone flap recommended by the TBI guideline [11], which also strongly emphasizes the sufficient temporobasal decompression. In the contemporary literature, some authors already postulate craniectomies overreaching the 180 cm^2^ proposed by the TBI guideline [9,15]. These authors emphasize that larger DC size positively impacts both approach-related surgical complications and long term outcome in patients with traumatic brain injury, spontaneous intracerebral hemorrhage, and cerebral infarction [9,14,15,16,17]. Our findings support the benefit of the larger bone flap in poor-grade SAH patients undergoing DC. Similar to the results of Jabbarli et al. [17], our data showed no difference in long term outcome between the groups but showed a trend of lower mortality rates in patients with larger DC size. The lack of statistical significance in our outcomes might be attributed to generally large size of the DC in our cohort, where both groups exceeded the 105 cm^2^ proposed by Jabbarli et al. [17]. Thus, the severity of the underlying SAH seems to be the main factor for long-term functional outcome [18].

The DC is known to significantly reduce short term mortality in patients with SAH [18]. The size of the DC of more than 105 cm^2^ proposed by Jabarli et al. [17] seems to also reduce risk of infarction and reduce the length of postoperative ICP treatment in SAH patients. Our study supports the thesis that larger craniotomy also seems to reduce the risk of approach-related sheer bleeding more effectively. Therefore, 150 cm^2^ might offer even better condition for brain swelling to surpass and prevent the tissue damage. Contrary to Sedney et al. [14], we did not find any long term complications of the larger bone flap. However, there are conflicting results on the positive effect of the DC on both short- and long-term outcome [3,18,19,20]. Ongoing prospective PICASSO Trial [6] should give the definitive answer on this question.

The analysis of postoperative CT scans showed that 13/14 shear-bleeding lesions were localized dorsomedially. With only one bleeding located temporobasaly, which supports the idea of routinely performed temporobasal decompression. All these findings support larger extension of planned craniectomy, as shown in Figure 1.

Due to the retrospective design of the study, we were not able to differentiate between the primary planed decompressive craniectomy and secondary intraoperative decision of bone flap ommitance. However, in case of patients with SAH WFNS-grade 4 and 5, careful preoperative approach planning with larger craniotomy size seems to be rational to be eventually prepared for intraoperative decisions for the bone flap ommitance.

Although not routinely used in our clinical setting, we retrospectively assessed the PRESSURE score, which was proposed by Jabbarli et al. [2] to evaluate the necessity of DC in patients with SAH. As expected, our cohort showed generally higher PRESSURE scores with a median of 6 points (range 0–9 points). According to the published study, 64% of patients who reached 6 points and above required the DC. This suggest that the PRESSURE score has potential clinical value. However, a more detailed evaluation of the score goes beyond the scope of this study.

In the literature, there are contradictory statements to the exact size of the DC. In his retrospective study, Schimmer et al. [8] found a clear correlation between the size of the DC and the effectiveness ICP reduction. Another retrospective cohort by Schur et al. also suggests that the larger size of the craniectomy provides better odds for controlling the elevated ICP [21]. However, Holsgrove at al. [7] saw no correlation between the size of the DC and long term outcome.

In case of secondary ommitance of the bone flap resulting in small DC, the size of the bony decompression might not suffice to control the elevated ICP. In such cases, second surgery and DC enlargement may be necessary [12]. Furthermore, the presence of cerebral vasospasm and delayed cerebral ischemia might cause secondary elevation of the ICP in course of the therapy [22] and subsequently developed infarction might negatively influence the outcome [19]. In the acute phase of the SAH, the underlying brain swelling is a major factor influencing morbidity and mortality [6]. Larger DC size might potentially prevent secondary ICP elevation and subsequent severe cerebral herniation.

In case of hemorrhagic progression of shear-bleeding lesion, a demonstrated mechanism of approach-related injuries could result in development of ipsilateral postoperative hematoma. In such cases, revision surgery might also be necessary due to the mass effect of the hemorrhage [23,24]. In order to prevent these complications, careful preoperative planning and a larger extent of the surgical approach should provide better ICP control and prevent unnecessary hemorrhagic complications.

According to the contemporary literature, the timing of the DC is crucial. Generally, primary decompression performed within 2 days after aneurysm rupture is favored by most authors [5,8,25,26]. In 2006, Smith et al. proposed prophylactic DC for selected patients in order to avoid all potentially fatal complications of SAH therapy, aiming to improve both short- and long-term outcome in SAH patients [3]. In our cohort, we did not find any patients who underwent late decompressive craniectomy (<48 h).

Contrary to the proposed theory of higher incidence of shunt-dependent hydrocephalus by patients who undergo larger decompressive craniectomy [27], our cohort showed same rates of hydrocephalus in both groups. The limitation of this statement is the low number of patients in the group without shear bleeding.

The main limitations of the presented study is the scarcity of the SAH, resulting in small numbers in predefined subgroups. Furthermore, the retrospective design of the study does not allow us to define the exact decision time for the DC (preoperatively/intraoperatively). Therefore, we were not able to differentiate between primary DC and ommitance of the bone flap reinsertion. Finally, the number of patients suffering approach-related complications is relatively small, which limits our statistical evaluation of the outcome.

## 5. Conclusions

Our findings suggest, that for poor-grade SAH patients indicated for surgical clipping, a larger bone flap should be planned in advance in order to reach a sufficient decompression if needed. Because most of the shear-bleeding lesions were localized dorsomedially, we suggest the dorsal margin of the DC should be placed close to the lambdoid suture (≥2 cm). Such extension of the craniectomy should provide the necessary surface to effectively reduce the ICP and to prevent approach-related complications.

## Figures and Tables

**Figure 1 brainsci-13-00371-f001:**
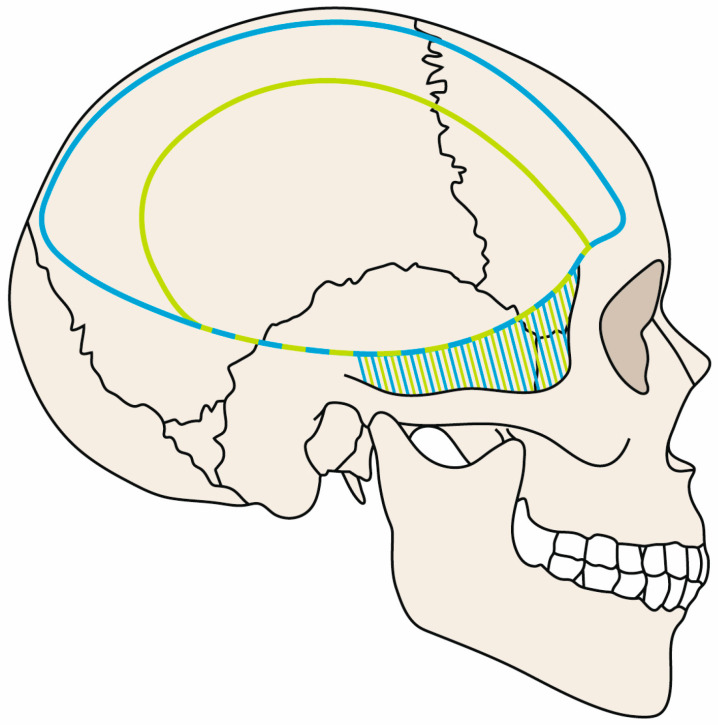
Size of the craniectomy—blue line demonstrates the craniectomy margin for larger DCs. Green line demonstrates the craniectomy margin for smaller DCs. Dashed green blue line represents the common temporobasal margin, and green blue area demonstrates common osteoclastic temporobasal decompression.

**Figure 2 brainsci-13-00371-f002:**
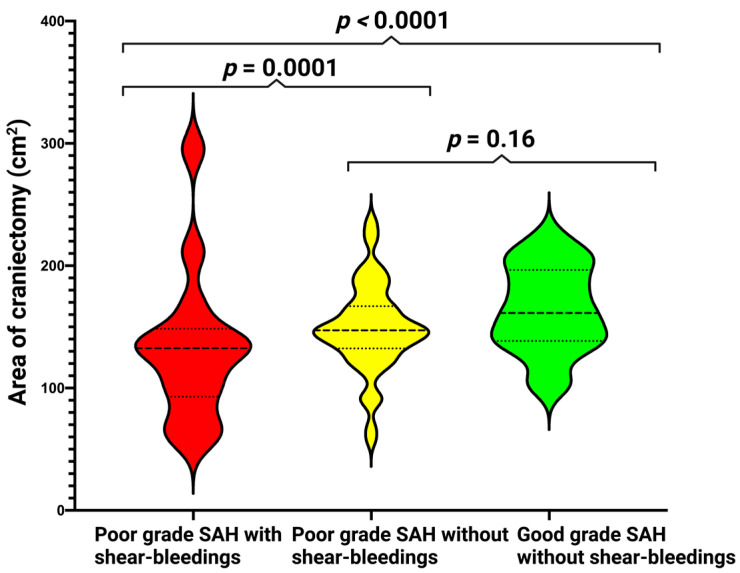
Violin plots shows mean and distribution of area of craniectomy in patients stratified by: RED—poor-grade SAH with shear bleedings, YELLOW—poor-grade SAH without shear bleedings, GREEN—good-grade SAH without shear bleedings. The thick dashed horizontal lines in the violin plots constitutes the median values, dotted lines in violin plots represent interquartile range, and *p* values represent the results of Student’s *t* test; SAH—subarachnoid hemorrhage.

**Figure 3 brainsci-13-00371-f003:**
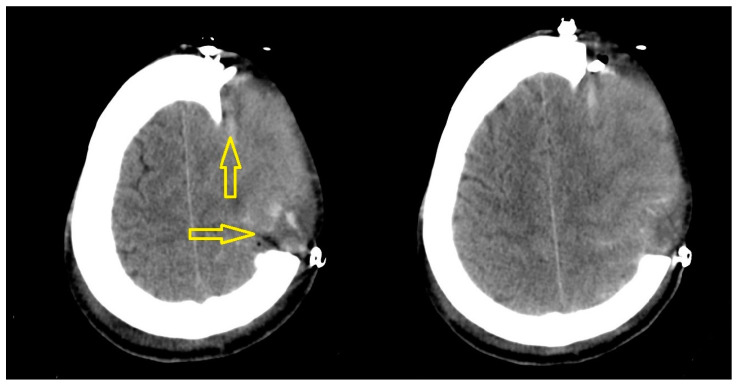
Postoperative CT scan demonstrating shear bleeding lesions with underlying cerebral herniation.

**Table 1 brainsci-13-00371-t001:** Patient characteristics table.

Patient Characteristic	*n*
Male	28
Female	60
WFNS 2	10
WFNS 3	11
WFNS 4	16
WFNS 5	51
Fisher-grade 2	2
Fisher-grade 3	82
Fisher-grade 4	4
Mean age (±MD) in years	55.04 ± 13.53
Shear bleeding complications	14

WFNS—World Federation of Neurological Societies Subarachnoid Hemorrhage Grading, MD—mean deviation.

**Table 2 brainsci-13-00371-t002:** Good-grade vs. poor-grade comorbidities.

	Good-Grade SAH (*n* = 21)	Poor-Grade SAH (*n* = 67)	*p* Values
Sex (Male:Female)	10:11	18:49	0.1
Arterial hypertension	9	24	0.6
Smoking	9	26	0.8
Preoperative blood thinners	3	16	0.5
PRESSURE score (IQR)	6 (4.5–7)	6 (5–7)	
Aneurysm localization:			
Anterior cerebral artery (A1 segment)	1	2	
Anterior cerebral artery (A2 segment)	0	1	
Anterior cerebral artery (A3–A4 segment)	0	2	
Anterior communicating artery	5	13	
Internal carotid artery—ophthalmic segment	0	5	
Internal carotid artery—terminal segment	2	12	
Medial cerebral artery—bifurcation	9	33	
Basilar artery—terminal segment	0	1	
Posterior cerebellar artery	1	1	
Aneurysm size (mm)	11 ± 7.3	11 ± 7.7	

SAH—subarachnoid hemorrhage, IQR—interquartile rage.

**Table 3 brainsci-13-00371-t003:** Poor-grade SAH with and without shear bleedings—comparison of outcomes.

	Poor-Grade SAH with Shear Bleedings (*n* = 14)	Poor-Grade SAH without Shear Bleedings (*n* = 53)	*p* Value
mRS < 4 in 6 months	12	8	0.99
1-year mortality	10	23	0.07
VP shunt implantations	2	18	0.2

SAH—subarachnoid hemorrhage, mRS—modified Rankin Scale, VP shunt—ventriculo peritonealer shunt.

## Data Availability

The data are available on request from the author [M.V.] upon reasonable request.

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
