# Peer review of "Size of Craniectomy Predicts Approach-Related Shear Bleeding in Poor-Grade Subarachnoid Hemorrhage"

_brainsci, 2023, doi:10.3390/brainsci13030371_

Round 1

Reviewer 1 Report

This is a retrospective study assessing the correlation between the size of the bone flap and shear bleeding in patients with poor grade subarachnoid hemorrhage. Authors reported smaller size of the bone flap in patients with new shear-bleeding lesions on post-op CT scans compared to those without. No significant difference in mortality rates or hydrocephalus was observed. 

Comments:

- The baseline characteristics of the population studied are not sufficient (age, sex and WFS/Fisher) and further characteristics are needed. Please include the presence of medical comorbidities (include either the Charlson comorbidity index or APACHE score) to better assess outcomes, the length of ICU stay, the duration of mechanical ventilation (if needed), aneurysm location and size, signs of cerebral herniation (baseline pupil exam). 

- Please report the results of the functional outcome (mRS) in the abstract as this is important. No difference was seen between the two groups. 

- There is a contradiction between the results reported in the abstract versus what is mentioned in the results section. The p-value for secondary hydrocephalus is 0.2 in the abstract and 0.33 in the results, and the total number of patients without shear bleeding is different (45 vs 53). Please report the correct result.

- Please change the hydrocephalus outcome in the abstract to the need for VP shunt. A lot of SAH patients will develop hydrocephalus, but not all of them will need VP shunts. 

- Please revise the manuscript to improve punctuation and correct grammatical mistakes; there are quite a few of them

- Include at least 3 MeSH terms (keywords) as I do not see that. Suggested terms include subarachnoid hemorrhage, decompressive craniectomy, shear-bleeding ... etc

Reviewer 2 Report

I am glad to have the opportunity to review your work. This study was aimed to to analyze the size of the bone flap according to approach-related complications in patients with poor-grade SAH.

There are minor flaws, regarding English grammar which need to be corrected, as well as text formatting. Keywords need to be added.

The paper has similarities with previously published paper from one of the authors of the paper that is reviewed - https://www.ncbi.nlm.nih.gov/pmc/articles/PMC8338868/#__ffn_sectitle

However, the topic is clinically valuable. Figures are good.

Still, the introduction needs to be expanded. There needs to be more data presented on similar studies in the literature.

Statistical analyses is obscure. I suggest performing correlation analysis.

Also, PRESSURE score might have been used in order to predict primary and secondary DC.

Therefore, I recommend revision of the paper.

Reviewer 3 Report

1. According to the Materials and Methods and Results sections, patients with poor-grade SAH were included and the number is 88. However, the actual number of patients with poor-grade SAH (WFNS 4 and 5) is 67 (16 + 51) based on Table 1.

2. In the title, abstract, and introduction sections, poor grade SAH patients were the main research subjects. However, the authors focused on both patients with good-grade SAH and those with poor-grade SAH in the results and discussion sections.

3. To rule out confounding factors, the characteristics listed in Table 1 including gender and age should be compared between groups. In addition, other characteristics have not been included, such as hypertension, diabetes, coagulation disorders, etc.

4. In the Results section, “…102.1 ± 45.2 cm2 102 vs. 150.8 ± 37.43 cm2, p > 0.0001”. P >0.0001 means no significance between the two groups.

5. The outcomes between the groups including mRS, mortality, and VP shunts should be shown in a table.

6. The mean size of craniectomy in the PGn group was 150.8 ± 37.43 cm2, which could not reach the conclusion that “a surface of 150 cm2 with sufficient dorso-medial extension should be reached in order to prevent approach-related 208 complications, such as shear-bleeding lesions”.

Round 2

Reviewer 3 Report

The headline of Table 3 "Good-grade vs. poor-grade comorbidities" is not correct and should be revised.

Author Response

We revised the name of the Table 3 to following: "Poor-grade SAH with and without shear-bleedings – comparison of outcomes," as we reported on outcomes of poor-grade SAH.

Thank you for the proposition.

Sincerelly Yours

Martin Vychopen

(on behalf of all authors)